# May predator body-size hamper furtive predation strategy by aphidophagous insects?

**Roberto Meseguer**[1]*, **Alexandre Levi-Mourao**[1], **Marc Fournier**[2], **Xavier Pons**[1],
**Eric Lucas**[2]

1 Department of Crop and Forest Sciences, University of Lleida–Agrotecnio-CERCA Centre, Lleida, Spain,
2 Laboratoire de lutte biologique, Département des sciences biologiques, Université du Québec à Montréal,
Montréal, Canada

* roberto.meseguer@udl.cat

pone.0256991

TUNISIA

**Data Availability Statement:** All relevant data are
within the manuscript and its Supporting
Information files.

## Abstract

Furtive predation is an uncommon predation strategy within aphidophagous insects, as it
can be constrained by several factors. So far, the few reported furtive predators are characterized by their small body-size, vermiform shape, and slow movement. They live within the
aphid colonies, without triggering significant defensive acts, nor disrupting colony structure.
In this study, we aim to determine how body-size may prevent adoption of a furtive predation
strategy. For that, the American hoverfly, *Eupeodes americanus* (Wiedemann) (Diptera:
Syrphidae) was selected as a model species, according to the great body-size increase
experienced during the larval stage. We hypothesized that smaller instars will be furtive
predators, whereas larger ones will be active-searching predators. After the inoculation
close to a pea aphid, *Acyrthosiphon pisum* (Harris) (Hemiptera: Aphididae) colony, several
behavioral parameters of the different larval instars were recorded. The elicited aphid colony
disturbance was also evaluated and compared with that of the active-searching ladybird
beetle, *Harmonia axyridis* (Pallas) (Coleoptera: Coccinellidae), and of the furtive predator,
*Aphidoletes aphidimyza* (Rondani) (Diptera: Cecidomyiidae). Aphids showed significantly
fewer defensive behaviors in the presence of *E. americanus* larvae than in the presence of
the active-searching *H. axyridis*. Furthermore, our results clearly indicate that body-size
increase was not a limit, since the three larval instars of the American hoverfly acted as furtive predators, just like the furtive *A. aphidimyza*. It is the first time a furtive predatory behavior has been recorded on such a large aphidophagous predator. The obtained results
provide essential information about the biology of *E. americanus*, a potential biological control agent of aphids.

## 1. Introduction

Predation strategies greatly differ among insect species and critically affect their performance
when exploiting their prey. In aphidophagous predators, several species, such as ladybirds or
lacewing larvae, use an active-searching strategy that elicits an array of aphid defensive behaviors, which usually leads to a strong disruption of the aphid colony structure [1–3]. Some other

**Funding:** This work was supported by a grant from the Spanish Ministerio de Ciencia, Innovación y Universidades (project AGL2017-84127-R) and by a CRSNG discovery grant to EL. RM holds the predoctoral fellowship FPI-PRE2018-083602 from the Ministerio de Ciencia, Innovación y Universidades and ALM holds a predoctoral JADE Plus fellowship from the University of Lleida (Spain). The funders had no role in study design, data collection and analysis, decision to publish, or preparation of the manuscript.

**Competing interests:** The authors have declared that no competing interests exist.

predators, such as nabids, phymatids or mantids are considered ambush predators attacking mainly prey mobile stages [4]. Conversely, furtive predators such as the cecidomyiid midge *Aphidoletes aphidimyza* (Rondani) (Diptera: Cecidomyiidae) or the chamaemyiid *Leucopis annulipes* Zetterstedt (Diptera: Chamaemyiidae) live within the aphid colonies, without triggering defensive acts, nor disrupting their structure [3]. This leads to a higher amount of available food and increases their survival due to a dilution and a selfish herd effect, by choosing the central position of their aphid prey colonies, which reduces the probability of being attacked once the prey patch has been detected by other predators [3, 5, 6]. Furthermore, in ant-attended aphid colonies, furtive predation limits ant aggression and provides an enemy-free space for furtive predators against intraguild active-searching predators [7].

Among predatory insects, furtive predation is until now restricted to small species/organisms (3–4 mm max). Besides the searching behavior and the consumption rate, predator body-size is an important factor influencing aphid-predator interactions [1, 2, 5]. Implicitly, the larger the predator, the higher the perturbation, and consequently the level of defensive response by the colonial prey. Thus, within the same species, it is expected that older/larger instars, which move faster and create more leaf vibrations, trigger more defensive actions than younger/smaller ones. We formulate the hypothesis that such a combination of traits is incompatible with a furtive predation strategy.

Aphidophagous syrphids (Diptera: Syrphidae) are important biological pest control and pollination providers in agricultural systems [8, 9]. They lay eggs near the aphid colonies and show a prey density-dependent oviposition response [10–14]. Thus, after hatching, small mobility-reduced larvae can easily access food resources. Since 1st instar larvae are usually found within undisrupted aphid colonies [15], it can be presumed that their predation strategy is furtive, like that of *A. aphidimyza* or *L. annulipes*. However, despite a similar morphology, syrphid larvae undergo a significant increase in vigor, mobility, and body-size throughout their development [16, 17]. The body weight of *Eupeodes corollae* (Fabricius) increased from the 0.04 mg newly hatched 1st instar larvae to the 45 mg peak-weight 3rd instar [16]. In addition, changes in their chasing behavior are shown too, as 3rd instar larvae tend to raise up their head and rapidly and repeatedly strike forward and sideways, which is not described in furtive predators. As a model species, the American hoverfly, *Eupeodes americanus* (Wiedemann) was selected for the current study. It is a Nearctic predator widespread across North America [18] and a generalist aphid natural enemy [19]. Since a short while ago, it is under evaluation as a new commercial biocontrol agent in Canada. However, aside from its good performance shown at low temperatures [20, 21], little is known about the ecology of *E. americanus*.

In this study, we aim to determine how body-size increase could conflict with the adoption of a furtive predation strategy by aphidophagous predators. We then formulated the hypothesis that predator's body-size would be a limiting factor to furtive predation. We predict that: (i) younger/smaller 1st instar larvae of *E. americanus* will act as furtive predators, whereas older/larger instars (2nd and 3rd) will act as active-searching predators; and (ii) an increase in the body-size will have a significant impact on larval performance when exploiting aphid colonies, triggering more defensive responses and disrupting aphid colony cohesion (hampering furtive strategy adoption).

In order to test the hypothesis, we compared: (i) the aphid defensive acts triggered by the 1st, 2nd and 3rd instar larvae of *E. americanus* with those of the active-searching predator *Harmonia axyridis* (Pallas) (Coleoptera: Coccinellidae), and with large larvae of the furtive predator *A. aphidimyza*; and (ii) the aphid defensive acts triggered by each larval instar, within the same predatory species, when inoculated near an *Acyrthosiphon pisum* (Harris) (Hemiptera: Aphididae) colony.

## 2. Material and methods

### 2.1. Insect rearing

**2.1.1. Aphids.**   *Acyrthosiphon pisum* was reared on broad bean (*Vicia faba* L.) plants inside plastic framed cages (35 x 35 x 35 cm) covered with muslin.

**2.1.2. Predators.**   Adult individuals of *E. americanus* were collected in the experimental fields of the Centre de Recherche Agroalimentaire de Mirabel (CRAM; Mirabel, Quebec, Canada). They were carried to the Laboratory of Biological Control at Université du Québec à Montréal (UQAM), and placed into rectangular nylon mesh cages (81 x 53 x 60 cm). Adults were fed with a sugar water mixture and with an artificial flower consisting of a round cotton makeup remover saturated with wildflower honey diluted with water (1:2 w/v) and covered with wildflower bee pollen. One broad bean plant infested with *A. pisum* was provided as an oviposition substrate. The plant was replaced three times per week and syrphid eggs were collected and placed into separate plastic boxes (25 x 15 x 8.5 cm) according to the collection date. Once hatched, larvae were fed *ad libitum* with a diet of *A. pisum*. Aphids were provided on the leaflets of a single broad bean stem, which was placed into a filled water glass vial sealed with parafilm to prevent dehydration.

The harlequin ladybird beetles, *H. axyridis*, were collected from an overwintering population in Sainte-Agathe-de-Lotbinière (Quebec, Canada) and carried to the Laboratory of Biological Control at UQAM. A stock culture was maintained and refreshed yearly with field-captured individuals. Adults were kept in plastic boxes (25 x 15 x 8.5 cm) and fed *ad libitum* with *A. pisum*, decapsulated cysts of *Artemia franciscana* (Kellog) (Anostraca: Artemiidae), wildflower honey diluted with water and wildflower bee pollen. Aphids were provided on the leaflets of broad bean stems using the same procedure as above. A crumpled piece of towel paper was provided as an oviposition substrate. Three times per week, eggs were collected and placed into Petri dishes on humid cotton. After hatching, larvae were carefully moved to separate plastic boxes according to the hatching date with the same diet as adults.

The aphid midge *A. aphidimyza* was purchased from the commercial supplier Anatis Bio-protection (Saint-Jacques-Le-Mineur, Quebec, Canada). Midges were reared inside plastic framed cages (35 x 35 x 35 cm) covered by muslin. Several broad bean plants infested with *A. pisum* were provided as food and oviposition substrate. New plants were added once a week and the older ones were discarded. Three times per week eggs were collected and placed into separate plastic boxes (25 x 15 x 8.5 cm) according to the collection date. Once hatched, larvae were fed *ad libitum* with *A. pisum*, provided on broad bean stem leaflets following the same procedure as above.

Aphid and predator populations were reared in a growth chamber under optimal conditions: 24°C ± 1°C, 75% ± 5% RH and a light regime of 16L:8D photoperiod.

### 2.2. Evaluation of the predatory strategy of *E. americanus* and effects of the predator body-size on aphid colony's disturbance

*Acyrthosiphon pisum* was selected for this experiment since this species shows significant defensive behaviors (see below), which can be easily observed [22]. Tests were carried out on 20–40 cm height broad bean plants at 23–25°C, RH 24–30%, 16L: 8D photoperiod. One different plant was used per test. On each experimental plant, the abaxial side of a leaflet was infested with 2nd, 3rd and 4th instar nymphs enclosed in a clip-cage (4 cm diameter, 2 cm height). After 24 h, clip-cages were removed and aphid colonies were carefully standardized to 13–16 individuals.

After a ten-minute delay, one predator per plant was released near the petiole of the aphid-infested leaf, marking the beginning of the experiment. Aphid defensive acts were then recorded by direct observation during the following 45 min or until the predator left the leaflet. Aphid defensive acts were classified as follows: (1) **Walking away** from the feeding site, (2) **Dropping** off the plant, (3) **Kicking** and (4) **Wriggling**. The length of the visit, number of consumed aphids, and aphids left on the leaf after the visit were recorded. For each predator, the attack success was evaluated in terms of the number of prey consumed / the number of contacts with prey (number of prey consumed + number of contacts that triggered defensive actions). Predators that did not interact with the aphid colonies were discarded and replaced.

To characterize the predation strategy of *E. americanus* larvae, 8 experimental treatments were tested:

A control treatment without predators

i.   *aphidimyza* 2–3 mm old larvae (as a furtive predator model)

ii.  *H. axyridis* 1st instar larvae (as an active-searching predator model)

iii. *H. axyridis* 2nd instar larvae (as an active-searching predator model)

iv.  *H. axyridis* 3rd instar larvae (as an active-searching predator model)

v.   *E. americanus* 1st instar larvae (body-length $\approx$ 2mm) (strategy unknown)

vi.  *E. americanus* 2nd instar larvae (body-length $\approx$ 4mm) (strategy unknown)

vii. *E. americanus* 3rd instar larvae (body-length $\approx$ 8mm) (strategy unknown)

Each treatment was replicated 20 times. As there is still controversy regarding the number of *A. aphidimyza* larval instars (3 or 4) [23], we only used the largest larvae (2–3 mm), which have a similar body-size to the 1st instar larvae of *H. axyridis* and *E. americanus*. To standardize the level of hunger among larvae, medium and large predators (2nd and 3rd instar larvae) were subjected to starvation for a period of 24 h. To this effect, larvae were kept individually in a holding cage with a water-moistened dental cotton roll to prevent dehydration. Since starvation led to a high mortality of small larvae, only newly hatched individuals of *H. axyridis* and *E. americanus* were used in 1st instar larvae tests. No starvation was imposed on *A. aphidimyza* larvae for the same reason.

## 2.3. Statistical analysis

As aphid colony size varied from 13 to 16 individuals, the number of consumed aphids and aphids left after the predator's visit were expressed in percentage (by dividing data by the initial number of individuals in the colony and multiplying by 100). The number of recorded defensive acts was also standardized in two different ways: by dividing it (i) by the initial number of aphid individuals in the colony and (ii) by the length of the test (minutes), as this parameter was also important to determine the level of disturbance elicited on the aphid colony.

In order to eliminate the potential effect of predator body-size on the visit length, voracity, and attack success, comparisons were made between similar body-sized instars of the different predatory species (small, medium and large instars). Similar criteria were followed to determine the predation strategy of *E. americanus* larvae; only similar body-sized developmental instars of *H. axyridis* and *E. americanus* were selected for the comparison of the number of the defensive acts recorded. However, *A. aphidimyza* was considered in this analysis, regardless of its body-size, as it is our furtive predator model. To assess the effect of the predator body-size over the above mentioned parameters, the different instars within the same species were then compared. Data normality and variance homogeneity were evaluated respectively by

Kolmogorov-Smirnov and Levene's tests. Since data were not normally distributed, nonparametric Kruskal-Wallis tests were used to compare means. When significant differences were found, multiple comparisons were performed with pairwise Dunn tests applying the Bonferroni correction. All the statistical analyses were performed using the statistical software SPSS v. 25.0 (IBM Corp, 2017).

# 3. Results

## 3.1. Predatory behavior

### 3.1.1. Predatory species behavior.
Predator visits lasted, on average, $36.24 \pm 1.27$ min, ranging from 15 to 45 min (Fig 1A). The active-searching *H. axyridis* 3rd instar larvae showed the lowest visit length ($15.41$ min $\pm 3.29$) ($H = 55.15$, df = 7, $P < 0.001$). According to the results shown in Fig 1A, the visit length recorded within small and large body-sized instars was significantly different (small: $H = 12.57$, df = 2, $P = 0.002$; large: $U = 55.00$, df = 1, $P < 0.001$). *Eupeodes americanus* 1st instar larvae visit length did not differ significantly from that of *A. aphidimyza* (all tests lasted the maximum time allowed (45 min)), whereas *H. axyridis* 1st and 3rd instar larvae spent significantly less time on the infested leaf. No differences were recorded between medium instars ($U = 180.50$; df = 1; $P = 0.602$).

Relative to voracity, the American hoverfly 3rd instar larvae were, by far, the most voracious predators ($H = 43.23$, df = 7, $P < 0.001$), eating around 30% of the aphids in the colony. When comparing similar body-sized instars of the different predatory species, only significant differences were shown between large larvae ($U = 58.50$, df = 1, $P < 0.001$) (Fig 1B).

Although the attack success shown by *H. axyridis* 1st instar larvae was apparently lower than those recorded for *A. aphidimyza* and *E. americanus* 1st instar larvae, no significant differences were reported for any of the comparisons (small: $H = 3.77$, df = 2, $P = 0.152$; medium: $U = 166.00$, df = 1, $P = 0.696$; large: $U = 170.50$, df = 1, $P = 0.545$) (Fig 1C).

### 3.1.2. Influence of the body-size on the predator behavior.
A decreasing trend in the visit length was observed throughout the different instars of *E. americanus* ($45.00$, $41.11$ and $38.04$ min for the 1st, 2nd and 3rd instar, respectively), but no significant differences were recorded ($H = 5.44$, df = 2, $P = 0.07$). However, voracity significantly increased with larval body-size, reaching its maximum value at the 3rd instar ($4.83$, $7.36$ and $30.09\%$ for the 1st, 2nd and 3rd instar, respectively) ($H = 29.45$, df = 2, $P < 0.001$). Although the 3rd instar larvae showed the highest attack success, no significant differences were observed between the three larval instars of the American hoverfly ($73.81$, $71.76$ and $92.58\%$ for the 1st, 2nd and 3rd instar, respectively) ($H = 2.11$, df = 2, $P = 0.35$).

Regarding the active-searching *H. axyridis*, the visit length was significantly lower for the 3rd instar larvae ($32.94$, $38.59$ and $15.41$ min for the 1st, 2nd and 3rd instar, respectively) ($H = 19.18$, df = 2, $P < 0.001$). No differences in voracity were shown between larval instars ($4.83$, $8.81$ and $8.90\%$ for the 1st, 2nd and 3rd instar, respectively) ($H = 5.13$, df = 2, $P = 0.08$). However, the attack success significantly increased from the 1st to the 3rd instar ($48.61$, $69.94$ and $88.54\%$ for the 1st, 2nd and 3rd instar, respectively) ($H = 10.58$, df = 2, $P = 0.005$), when it reached its maximum value.

## 3.2. Impact on aphids

### 3.2.1. Aphid defensive response.
The number of defensive acts per aphid was significantly higher in the presence of the active searching *H. axyridis* larvae than in all the other treatments ($H = 19.67$, df = 3, $P < 0.001$ for 1st instar larvae; $H = 43.80$, df = 3, $P < 0.001$ for 2nd instar larvae and $H = 32.66$, df = 3, $P < 0.001$ for 3rd instar larvae), which were not different from one to another ($P > 0.05$) (Fig 2A). When considering the number of defensive acts per

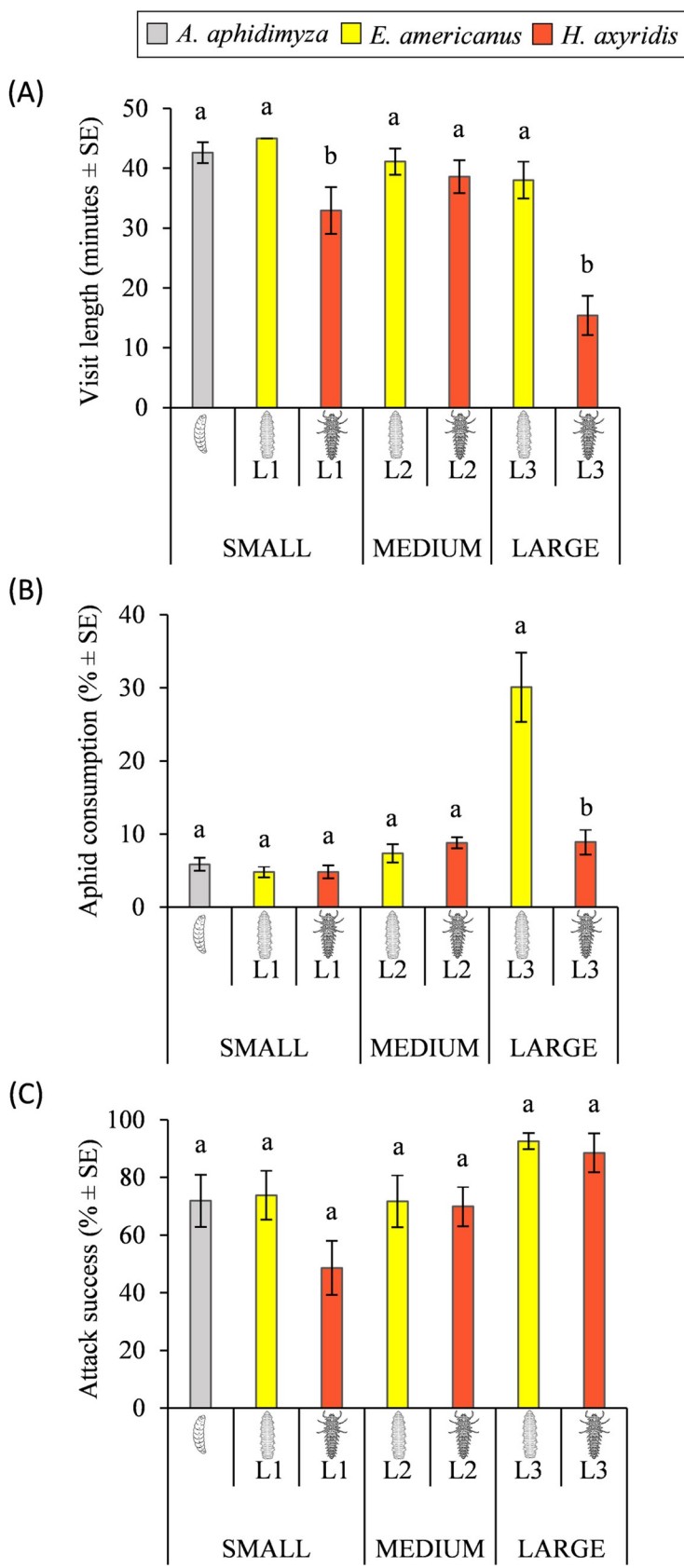

**Fig 1.** (A) Visit length (minutes ± SE), (B) aphid consumption (% ± SE) and (C) attack success (% ± SE) for each one of the different treatments (n = 20). Comparisons are made between similar body-sized instars (small, medium and large) of *Aphidoletes aphidimyza*, *Eupeodes americanus* and *Harmonia axyridis*. Test maximum length = 45 minutes. Different letters indicate significant differences ($P < 0.05$).

minute (see Fig 1A for visit length), treatments with *H. axyridis* larvae were those that elicited the highest amount too ($H = 34.75$, df = 3, $P < 0.001$ for 1st instar larvae; $H = 43.97$, df = 3, $P < 0.001$ for 2nd instar larvae and $H = 43.72$, df = 3, $P < 0.001$ for 3rd instar larvae), whereas no significant differences were shown between the rest of the treatments ($P > 0.05$) (Fig 2B).

Within species, no differences between the different larval instars were recorded regarding the number of defensive acts per aphid (Fig 2A). However, when considering the number of defensive acts per minute, our results show that *H. axyridis* 3rd instar larvae triggered significantly more defensive acts than the smaller instars ($H = 15.21$, df = 2, $P < 0.001$), while no differences were found for *E. americanus* larvae (Fig 2B).

Regarding the number of the different defensive acts recorded for each treatment (Table 1), the active searching predator, *H. axyridis* was the species that triggered the largest number of *droppings*, whereas more *walking away* was recorded in treatments with the furtive *A. aphidimyza* and *E. americanus*. Defensive acts such as *wriggling* and *kicking* were more frequently recorded in the control and *A. aphidimyza* treatments, where a similar number of the different defensive acts was recorded. No droppings were recorded in none of these treatments. Focusing on the number of *droppings*, our results show a continual increase through the different developmental stages of *H. axyridis* (from the 1st to the 3rd instar), whereas it remained equal for *E. americanus*, independently of the larval instar.

**3.2.2. Colony cohesion.** Regarding the number of aphids left after the predator's visit, treatments with *H. axyridis* 2nd and 3rd instar larvae were those which recorded the greatest impact over the aphid colony cohesion, respectively leaving 40 ± 6% and 28 ± 7% of the initial number of aphids in the colony. Conversely to the 1st and 2nd instar, *E. americanus* 3rd instar larvae left a significantly lower number of aphids than *A. aphidimyza* ($H = 26.84$, df = 3, $P < 0.001$). Our results show that, within species, larger instars left a significantly lower number of aphids than smaller ones ($H = 96.13$, df = 7, $P < 0.001$) (Fig 3).

## 4. Discussion

Predator body-size and predator/prey relative body-size are key factors modulating the behavior of the former. For example, prey preference changes with body-size [24–27]. In active-searching organisms, body-size of the predator and their prey are usually correlated. Conversely, ambush predators may have no direct relation between their body-size and that of their prey [28]. Also, colonial predators, such as army or legionary ants may attack huge prey and their capacity seems limited not by body-size but by colony size [29, 30]. For furtive predators, their strategy is based on a furtive way of living, being concealed within their prey colony, so that colony cohesion should be maintained (see [1, 5, 7]). Therefore, body-size should intuitively constitute a limit to the efficacy of the strategy.

However, contrary to our hypothesis, our results clearly demonstrate that the frequency of prey defensive acts, which is the way to characterize a furtive predator [3, 5], was not significantly different for any syrphid larval instar from the furtive Cecidomyiidae. It means that, despite a body-size (mm) increase of more than 100% between 1st and 2nd, and 300% between 1st and 3rd instar larvae of the American hoverfly, all the larval instars were clearly furtive predators. Thus, within the size range of syrphid larval instars, body-size is not a constraint. It is the first time that a furtive predatory behavior has been recorded on such a large predator.

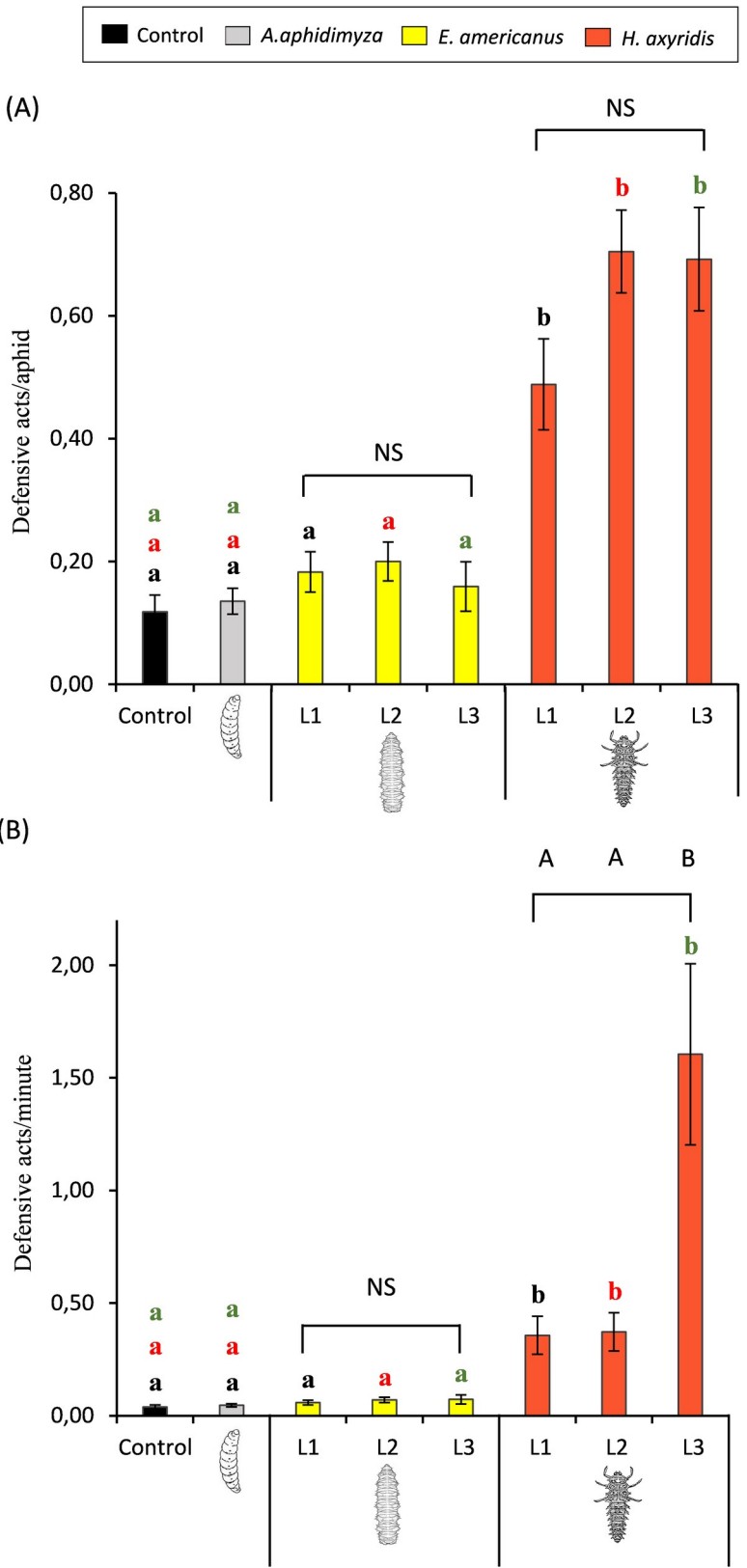

**Fig 2.** Average number of (A) defensive acts/aphid and (B) defensive acts/minute recorded for each treatment. Comparisons have been made between similar body-sized instars (small instars: black letters; medium instars: red

letters; and large instars: green letters) of *Aphidoletes aphidimyza*, *Eupeodes americanus* and *Harmonia axyridis* and between different instars within the same species (results are shown over the square brackets). Columns followed by different letters are significantly different at $P < 0.05$.

Furtive predation behavior has an adaptive value. First, by being unnoticed within a prey colony, the furtive predator gains easy access to nutritional resources. For example, neonate larvae of *A. aphidimyza* will starve to death if they are $> 63$ mm from food [31]. Second, for a small larval predator, searching for prey increases intraguild encounter probability and subsequent mortality risk [32]. By staying within the prey colony, it reduces the risk of predation. Third, while small instar larvae suffer from intraguild predation by other aphidophagous predators [17, 33, 34], Lucas and Brodeur [3] demonstrated that intraguild predation upon furtive predators is significantly reduced by living within high-density aphid colonies, through usurpation of aphids' defensive strategies such as the dilution and selfish herd effects. However, this probably depends on the nature of the intraguild predator species since, for instance, *A. aphidimyza* eggs are highly vulnerable for predation by predatory mites even if they are laid within aphid colonies [35]. Fourth, aphid-tending ants are significantly more aggressive on active-searching predators than on furtive predators (see [7]).

The behavior of active-searching ladybeetles and furtive Syrphidae were quite different. First, the visit length of furtive predators should exceed that of the more mobile active-searching predators. However, in the present study, limitations to 45 min of observation reduced the probability of finding significant differences. The larger active-searching predators, ladybeetle 3rd instar larvae, spent significantly less time on the infested leaf, probably due to its high mobility. However, the visit length recorded for the smaller ladybird larval instars (1st and 2nd instar) was probably impacted by the handling time, as they needed a long time to entirely consume one single prey and start moving again. This could explain the lack of significant differences between the results recorded for the ladybird and syrphid 2nd instar larvae. The velocity of the two types of predators also differs in the literature. Active-searching predators, such as ladybirds, present higher foraging ratios when compared with furtive predators. Frazer and Gilbert [36], recorded speeds up to 16.8 cm/min for adult coccinellids while Scott [37] found that the foraging rate of a 3rd instar syrphid larvae ranged from 1.6 to 3.8 cm/min.

Comparing voracity between similar body-sized larvae of the different predatory species, only significant differences were shown in the 3rd instar. This is a direct consequence of the different predation strategy used by each of the species. The large amount of aphid *droppings* elicited by the 3rd instar larvae of the ladybeetle, led to a low prey availability for consumption and thus, to a low voracity. This could also explain the lack of significant differences between

**Table 1. Cumulative total number of the different aphid defensive acts elicited by the different predators.**

| Predator | n | Walking away | Dropping | Wriggling | Kicking | Total |
|---|---|---|---|---|---|---|
| Control | 20 | 20 | 0 | 10 | 5 | 35 |
| *A. aphidimyza* | 20 | 21 | 0 | 11 | 8 | 40 |
| *E. americanus* (L1) | 20 | 26 | 19 | 7 | 1 | 53 |
| *E. americanus* (L2) | 20 | 27 | 19 | 6 | 5 | 57 |
| *E. americanus* (L3) | 20 | 26 | 16 | 4 | 2 | 48 |
| *H. axyridis* (L1) | 20 | 63 | 56 | 6 | 16 | 141 |
| *H. axyridis* (L2) | 20 | 53 | 141 | 5 | 9 | 208 |
| *H. axyridis* (L3) | 20 | 20 | 179 | 2 | 3 | 204 |

L1: 1st instar larvae; L2: 2nd instar larvae; L3: 3rd instar larvae.

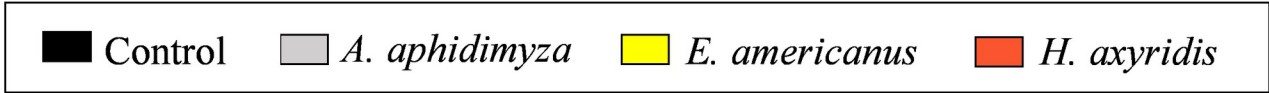

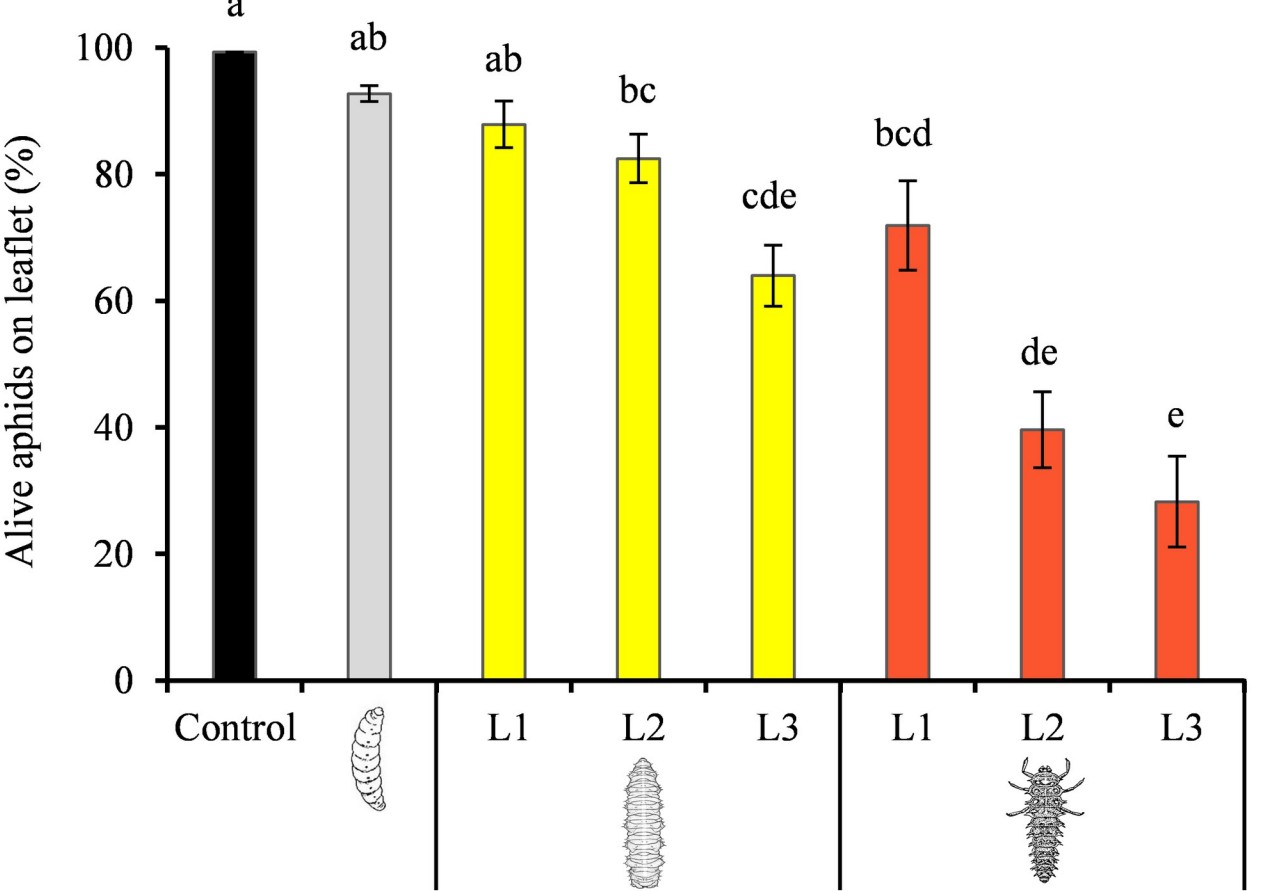

**Fig 3. Percentage of alive aphids present on the colony site (leaflet) by the end of the tests with *Aphidoletes aphidimyza*, *Eupeodes americanus* and *Harmonia axyridis* larvae (n = 20).** Columns followed by different letters are significantly different at *P* < 0.05.

the 2nd and 3rd instar ladybird larvae. Regarding the American hoverfly, a considerable increase between the 2nd and 3rd larval instar was recorded. This is probably bound to the great body-size increase experienced during the last larval instar, which is 100% bigger than the previous one, its lower handling time and high attack success. Nevertheless, even feeding upon 30% of the aphids in the colony, the American hoverfly 3rd instar larvae remained furtive.

Relative to the attack success, no differences were shown between treatments. The low value recorded for the 1st instar ladybird larvae is consistent with the one recorded by Fréchette et al. [5], which confirms the poor performance of small active-searching predators. However, this parameter is not a key factor for the predation strategy determination, since Lucas and Brodeur [3] recorded a similar attack success for *C. rufilabris* and *A. aphidimyza*, even though they are active-searching and furtive predators respectively.

Aphid's defensive response changes according to the predation strategy used by the natural enemy. Drastic differences were shown regarding the number of elicited aphid defensive acts.

The high foraging ratio of *H. axyridis* larvae and its campodeiform body-shape, possibly created more leaf vibrations than the slower and crawling *E. americanus* and *A. aphidimyza* larvae, triggering significantly more defensive acts by the aphids in the colony. Conversely, the latter two went unnoticed and barely disrupted the aphid colony. Our results are consistent with the study of Lucas and Brodeur [3], who observed that the active-searching predator *Chrysoperla rufilabris* Burmeister (Neuroptera: Chrysopidae), triggered significantly more aphid defensive acts than the furtive predator *A. aphidimyza*. In the same way, Fréchette et al. [5] recorded a higher disruption on the aphid colony structure when approached by *H. axyridis* 1st- 2nd instar larvae, than when approached by the furtive predators *L. annulipes* or *A. aphidimyza* larvae.

Contrary to our prediction, our results show that changes on *E. americanus* body-size, throughout the different larval instars, had no impact on the aphid colony disturbance. However, more studies need to be performed to confirm whether this pattern is also observed in other furtive predators and other aphid species. Relative to *H. axyridis*, the 3rd instar larvae triggered significantly more aphid defensive acts than smaller instars, implying that performance of active-searching predators can be heavily impacted by their body-size.

Regarding the different types of defensive behaviors expressed by aphids facing the different predators, our results show drastic differences. When facing active-searching predators, the aphids mainly responded by dropping off the plant, whereas more *walking away* was recorded in treatments with the furtive *A. aphidimyza* and *E. americanus*. Although some dropping was recorded for *E. americanus* larvae, this concurs with the results of Fréchette et al. [5], where the furtive predator *L. annulipes* also elicited some *drooping*, accounting for more than the 50% of the total number of elicited defensive acts, which is more than the proportion recorded for *E. americanus* in our study. They observed no differences in the proportions of aphid defensive behavior types elicited by *H. axyridis* 1st- 2nd instar larvae, *L. annulipes* and *A. aphidimyza* larvae, suggesting that aphids show similar behavior when reacting to active-searching and furtive predators. Nevertheless, they attribute this result to the fact that both *L. annulipes* and *A. aphidimyza* larvae triggered very few defensive acts, lowering the probability of finding a statistical difference between treatments. Brodsky and Barlow [38] demonstrated that *A. pisum* had a greater tendency to drop than walk away when approached by an adult of the ladybird *Adalia bipunctata* (L.) (Coleoptera: Coccinellidae), but showed the opposite tendency when approached by a larva of the syrphid *E. corollae*. They suggest that walking away may be an effective defensive act against slow moving furtive predators such as syrphid larvae. However, it may not be effective when approached by a fast moving active-searching predator, such as ladybird beetles, in which situation dropping could be the safest escape response. Chambers [39] also noted that the aphid *Aphis gossypii* Glover tended to move to upper surfaces of the leaves when they were followed by syrphid larvae of *E. corollae*. The numbers of the different elicited defensive acts remained almost equal for the different *E. americanus* larval instars, while for the active searching *H. axyridis*, the ratio *dropping*: *walking away* shown by *A. pisum* increased with the body-size of the approaching ladybird larvae. Dixon [1] observed a similar behavior when *Microlophium evansi* (Theobald) (Hemiptera: Aphididae) were approached by different instars of the ladybeetle *Adalia decempunctata* (L.) (Coleoptera: Coccinellidae). He attributed this behavior to a decrease of the ratio *speed of the aphid*: *speed of the predator*, making the walking away response of aphids less effective. He stressed too that different coccinellid instars with a similar speed of movement also elicited different defensive behavior patterns, suggesting that size may also play a role.

The survival of a furtive predator is intimately linked to the cohesion of the aphid colony. Aphids have to stay on the same site in sufficient numbers. Furtive predators, such as *A. aphidimyza* and *E. americanus*, barely triggered aphid defensive acts, causing low disturbance and

reducing the number of aphids in the colony by direct feeding on the prey. In our study, despite the high voracity recorded for the American hoverfly 3$^{rd}$ instar larvae, the aphid colony cohesion was better preserved than in treatments with 2$^{nd}$ and 3$^{rd}$ instar ladybird larvae. Conversely, the harlequin ladybird was the predator with the largest impact on the aphid colony cohesion. The low number of aphids left after the visits was a consequence of predator-induced defensive responses (mainly walking away and dropping) that sometimes have associated costs for the prey, such as loss of feeding or mating opportunities, predation by others predators etc. [22, 40–43]. In addition, by breaking this cohesion, active-searching predators can potentially access and kill furtive predators. Thus, from the biological control point of view, predation strategy may condition the way in which pest suppression is attained.

We may draw three main conclusions from the present study. First, that larvae of the American hoverfly, *E. americanus*, are clearly furtive predators, as they elicited a low number of aphid defensive acts, which was not significantly different from that of our furtive control, *A. aphidimyza*. Second, it appears that, contrary to our hypothesis, predator body-size is not necessarily a constraint to furtive predation strategy. Furthermore, the American hoverfly constitutes the largest furtive predator insect. An associated question is whether all, or most aphidophagous syrphid species (Syrphinae) (and most dipteran aphidophagous species) have a furtive predatory behavior; and whether the prey type (aphids, mites, adult whiteflies. . .) exploited may annul, reduce, or increase the adaptive value of a furtive predatory strategy. The furtive predation system implies a furtive predator and a gregarious prey which do not react strongly to the actions of the predator. Third, in an applied point of view, a key question for biocontrol is linked to the impact of the dropping behavior of the pest on the subsequent phytosanitary pressure on the crop. Is a high tendency to drop from the plant (in presence of active-searching predators) associated with a subsequent better situation, according to higher aphid mortality, or to a worse situation due to dispersion and creation of new colonies by dropped individuals? Another important issue would be to evaluate the potential of using an active-searching and a furtive predator combined in a biocontrol program.

## Supporting information

**S1 Fig.**
(TIF)

## Acknowledgments

Special thanks are due to the Centre de Recherche Agroalimentaire de Mirabel (CRAM; Mirabel, Quebec, Canada) for allowing sampling in their experimental fields. We also thank Marie D'Ottavio and Alice de Donder for the support with laboratory tests and Nathan Morris for English revision.

## Author Contributions

**Conceptualization:** Eric Lucas.

**Data curation:** Roberto Meseguer, Alexandre Levi-Mourao.

**Formal analysis:** Roberto Meseguer.

**Investigation:** Roberto Meseguer, Alexandre Levi-Mourao.

**Methodology:** Marc Fournier, Eric Lucas.

**Resources:** Xavier Pons, Eric Lucas.

**Supervision:** Xavier Pons, Eric Lucas.

**Validation:** Eric Lucas.

**Visualization:** Roberto Meseguer.

**Writing – original draft:** Roberto Meseguer.

**Writing – review & editing:** Alexandre Levi-Mourao, Marc Fournier, Xavier Pons, Eric Lucas.

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
