## [Decision Letter · Decision Letter 0]

5 Aug 2021

PONE-D-21-16778

May body-size hamper furtive predation strategy by aphidophagous predators?

PLOS ONE

Dear Dr. Meseguer,

Thank you for submitting your manuscript to PLOS ONE. After careful consideration, we feel that it has merit but does not fully meet PLOS ONE’s publication criteria as it currently stands. Therefore, we invite you to submit a revised version of the manuscript that addresses the points (minor revisions) raised during the review process.

We look forward to receiving your revised manuscript.

Kind regards,

Ramzi Mansour

Academic Editor

PLOS ONE

Journal Requirements:

This study was partially funded by the Spanish Government, Ministerio de Economia y Competitividad, under the framework of the research project ‘Arable crop management and landscape interactions for pest control’ (AGL2017-84127-R) and by a CRSNG discovery grant to Eric Lucas. Roberto Meseguer holds the predoctoral fellowship FPI-PRE2018-083602 from the Ministerio de Ciencia, Innovación y Universidades and Alexandre Levi-Mourao holds a predoctoral JADE Plus fellowship from the University of Lleida (Spain). 

This work was supported by a grant from the Spanish Ministerio de Ciencia, Innovación y Universidades (project AGL2017-84127-R) and by a CRSNG discovery grant to EL. RM holds the predoctoral fellowship FPI-PRE2018-083602 from the Ministerio de Ciencia, Innovación y Universidades and ALM holds a predoctoral JADE Plus fellowship from the University of Lleida (Spain).

Additional Editor Comments:

Reviewers' comments:

Reviewer's Responses to Questions

**Comments to the Author**

1. Is the manuscript technically sound, and do the data support the conclusions?

Reviewer #1: Yes

2. Has the statistical analysis been performed appropriately and rigorously? 

Reviewer #1: Yes

3. Have the authors made all data underlying the findings in their manuscript fully available?

Reviewer #1: Yes

4. Is the manuscript presented in an intelligible fashion and written in standard English?

Reviewer #1: Yes

5. Review Comments to the Author

Reviewer #1: This study provides new valuable knowledge about the predation behaviour of syrphid larvae compared to other aphidophagous predators. The whole study is all based om 45 minutes observations from single predators. The study is rather limited, but unless this providing quit a lot of data. The discussion is rather long and could be shortened a bit. Some more minor remarks are show below:

Title: Please change the title. Now it is not clear whether bode-size refers to the predator or prey.

Line 57: I think it is nice for the reader to shortly explain this hypothesis.

Line 85: “prevent adoption” is a bit awkward. Maybe better state: “could conflict with..

Line 122; What species of Artemia? What was the source? What kind of pollen and source? What kind of honey?

Line 136: in a climate cell?

Line 151-152: How exactly was aphid behaviour recorded. Per individual? Total number of kicks per colony? Was this corrected for the number of aphids?

156: would be nice to show these data as well

Line 160-170: what was the level of starvation of these predators. How did you standardize the willingness to search for food of these predators?

245: change “no” to “not”

Table 1: It is not clear what these numbers mean. For example walking away, does is mean that from each colony at least 1 aphid walked away? Or is this the total number of aphids that walk away from the 20 replicates with each 13-16 aphids? The experimental unit is one aphid colony, so here the response per colony should be presented.

Figure 1 is not clear. The resolution is too low. Also the small-medium and large classifications is now both in the x-axes and the legends. This is redundant. I think it is better to use the legend for species identification with different colours. Species should also be mentioned with full names in the caption.

Figure 2: The resolution is too low. Include full species names in the caption. Are these defensive acts based on the average acts per aphid per colony? So based on 20 replicates?

Figure 3: include full species names in the caption.

Line 311: depending on the stage I guess, please specify

Line 312-313: What means “looking for prey” . This statements also needs references

Line 317: Maybe in addition to this explain this may strongly depend on the predator species, as the study of

Messelink et al (Messelink, G. J., C. M. J. Bloemhard, J. A. Cortes, M. W. Sabelis, and A. Janssen. 2011. Hyperpredation by generalist predatory mites disrupts biological control of aphids by the aphidophagous gall midge Aphidoletes aphidimyza. Biological Control 57:246-252.), shows that aphidoletes eggs are highly vulnerable for predation by predatory mites and hiding within aphid colonies is not protecting them.

Line 338: Maybe include here these non-consumptive effects still can contribute to the control of aphids.

Line 356: was there also a difference observed in the production of alarm pheromones from the siphones by aphids when predated by ladybird larvae or the furtive larvae? Could this also be an explanation? Maybe good to include as well it might be more than just the leaf vibration.

Line 417 “from a fundamental point of view” can be omitted

Line 425-427: There is probably a lot of literature about this. So better refer to some other studies where they show the same mechanisms, or omit this discussion. I think it is a bit out of scope.

Line 428: Maybe mention here that the related species Eupeodes corollae is already on the market as BCA is Europe.

6. PLOS authors have the option to publish the peer review history of their article (what does this mean?). If published, this will include your full peer review and any attached files.

Reviewer #1: **Yes: **Gerben J. Messelink

---

## [Author Response · Author response to Decision Letter 0]

17 Aug 2021

Dear Dr. Mansour,

Thank you very much for giving us the opportunity of revising our manuscript after all these helpful comments and suggestions. We have gone over each point mentioned and made appropriate corrections to the initial manuscript. Here are the responses for each comment. In our responses, the referred line numbers correspond with the line numbering of the “Revised Manuscript with Track Changes.doc”.

Journal Requirements:

Done. We have checked PLO ONE’s style requirements and we have corrected little mistakes, such as eliminating the ZIP code and addresses from the affiliation information (see Title page), adding indentation at the beginning of each paragraph etc.

As we worked with four very common insect species, some of which are considered important agricultural pests (i.e. Acyrthosiphon pisum) or invasive species already settled in Canada (i.e. Harmonia axyridis), no licenses, permits nor institutional approvals were required to carry out this study. The full name of the authority that approved the field site access was already included on the text (see line 109) (Centre de Recherche Agroalimentaire de Mirabel (CRAM; Mirabel, Quebec, Canada)). 

3. Please remove any funding-related text from the manuscript and let us know how you would like to update your Funding Statement. 

Done. We have removed all the funding-related text from the acknowledgements section (see lines 457-463). 

Currently, your Funding Statement reads as follows: 

“This work was supported by a grant from the Spanish Ministerio de Ciencia, Innovación y Universidades (project AGL2017-84127-R) and by a CRSNG discovery grant to EL. RM holds the predoctoral fellowship FPI-PRE2018-083602 from the Ministerio de Ciencia, Innovación y Universidades and ALM holds a predoctoral JADE Plus fellowship from the University of Lleida (Spain).

We do agree with your Funding Statement proposal; it seems perfect to us. 

Additional Editor Comments:

Done. We have reviewed our reference list and now is correct. One reference was in a wrong style but it has been already corrected (see line 145). As we have added two new references (see lines 330 and 337) all the subsequent reference numbers have changed. 

Review Comments to the Author:

Title: Please change the title. Now it is not clear whether bode-size refers to the predator or prey. 

Done. We have modified the title to “May predator body-size hamper furtive predation strategy by aphidophagous insects?”

Line 57: I think it is nice for the reader to shortly explain this hypothesis.

Done. We have added a little explanation to what dilution and selfish herd effect are (see line 57-59). 

Line 85: “prevent adoption” is a bit awkward. Maybe better state: “could conflict with”.

Done (see line 87).

Line 122; What species of Artemia? What was the source? What kind of pollen and source? What kind of honey?

They were decapsulated cysts of Artemia franciscana. It was wildflower bee pollen and wildflower honey. This information has been added to the manuscript (see lines 113, 124-125).

Line 136: in a climate cell?

Yes. We added this information (see line 139).

Line 151-152: How exactly was aphid behaviour recorded. Per individual? Total number of kicks per colony? Was this corrected for the number of aphids?

The number of the different defensive acts was recorded per test, and thus, per colony. As we state in lines 183-185, data was standardized by dividing the number of defensive acts recorded on each replicate by the initial number of aphid in the colony.

Line 156: would be nice to show these data as well

We think this data is not relevant as it was only and exclusively used to calculate the predator attack success, so, in some way, this data is already reflected in this parameter. Furthermore, it would entail a big amount of data (20 replicates * 7 treatments = 140 rows). However, if you consider it essential, we can add a table with this data as supplementary material.

Line 160-170: what was the level of starvation of these predators. How did you standardize the willingness to search for food of these predators?

We added the following paragraph (see lines 174-179):

“To standardize the level of hunger among larvae, medium and large predators (2nd and 3rd instar larvae) were subjected to starvation for a period of 24 h. To this effect, larvae were kept individually in a holding cage with a water-moistened dental cotton roll to prevent dehydration. Since starvation led to a high mortality of small larvae, only newly hatched individuals of H. axyridis and E. americanus were used in 1st instar larvae tests. No starvation was imposed on A. aphidimyza larvae for the same reason.” 

Line 245: change “no” to “not”

Done (see line 255).

Table 1: It is not clear what these numbers mean. For example walking away, does is mean that from each colony at least 1 aphid walked away? Or is this the total number of aphids that walk away from the 20 replicates with each 13-16 aphids? The experimental unit is one aphid colony, so here the response per colony should be presented.

As stated in the caption, these numbers represent the cumulative total number of the different defensive acts recorded from the 20 replicates (i.e. if the same aphid walked away and then dropped, these defensive acts were considered as independent, thus, we registered 1 walking away + 1 dropping). With this table we just want to show to the reader the big differences we recorded in the aphid defensive behavior depending on which was the approaching predator. We respect the reviewer opinion but we think that if we present here the responses per colony (the mean of our replicates), differences can be underestimated. We think that, by presenting raw data (total number of the different defensive acts triggered by the different predators), the reader can better notice the big differences between treatments. 

Figure 1 is not clear. The resolution is too low. Also the small-medium and large classifications is now both in the x-axes and the legends. This is redundant. I think it is better to use the legend for species identification with different colours. Species should also be mentioned with full names in the caption.

Figures were sent as TIFF files, with high resolution, but by default, they appear like this after the PDF building. As reviewer suggested, we now use the legend for species identification with different colors. Species are now mentioned with full names in the caption (see lines 226-231).

Figure 2: The resolution is too low. Include full species names in the caption. Are these defensive acts based on the average acts per aphid per colony? So based on 20 replicates?

Figures were sent as TIFF files, with high resolution, but by default, they appear like this after the PDF building. As reviewer suggested, we have included full species names in the caption (see lines 267-274). Yes (on the upper panel). On the lower panel defensive acts are based on the average acts per minute per colony (or test).

Figure 3: include full species names in the caption.

Done (see line 302-305).

Line 311: depending on the stage I guess, please specify

It is already specified in the same sentence, as we refer to neonate larvae (see line 328).

Line 312-313: What means “looking for prey”. This statements also needs references

We have changed it to “searching for prey” (see line 329). We added a reference to justify the statement (see line 330).

Line 317: Maybe in addition to this explain this may strongly depend on the predator species, as the study of Messelink et al (Messelink, G. J., C. M. J. Bloemhard, J. A. Cortes, M. W. Sabelis, and A. Janssen. 2011. Hyperpredation by generalist predatory mites disrupts biological control of aphids by the aphidophagous gall midge Aphidoletes aphidimyza. Biological Control 57:246-252.), shows that aphidoletes eggs are highly vulnerable for predation by predatory mites and hiding within aphid colonies is not protecting them.

Done. We have added a sentence clarifying this with the suggested reference (see line 335-337).

Line 338: Maybe include here these non-consumptive effects still can contribute to the control of aphids.

This topic is already addressed in lines 427-430, when we talk about the aphid colony cohesion. 

Line 356: was there also a difference observed in the production of alarm pheromones from the siphones by aphids when predated by ladybird larvae or the furtive larvae? Could this also be an explanation? Maybe good to include as well it might be more than just the leaf vibration.

This parameter was not taken into account in our study, so we cannot answer to these questions. Anyway, we can assure that H. axyridis medium/large larvae triggered considerable leaf vibrations, which usually were followed by a big amount of aphid defensive acts. That is why we settle this hypothesis, based in what we observed. We appreciate the reviewer suggestion but we think that talking here about the alarm pheromone topic would be merely speculative. 

Line 417 “from a fundamental point of view” can be omitted

Done. We have omitted it (see line 437).

Line 425-427: There is probably a lot of literature about this. So better refer to some other studies where they show the same mechanisms, or omit this discussion. I think it is a bit out of scope.

We agree with the reviewer. This section has been omitted (see lines 444-447).

Line 428: Maybe mention here that the related species Eupeodes corollae is already on the market as BCA is Europe.

We agree with the reviewer but, with the aim of shortening the discussion section, we have omitted this section, as it is already mentioned in the introduction (see line 83-84).

Additional changes

Line 51: as in the first line we are talking about “insect species”, we have eliminated “crab spiders” from the examples.

Lines 164-167: the word “model” has been added to the description.

Line 402: the full name species, authorship, order and family have been eliminated, as this is not the first appearance in text. 

Line 448-449: to shorten the discussion section, we have omitted this sentence, as it is already mentioned in the introduction (see line 83-84). 

Acknowledgements: we have modified this section.

---

## [Editor Report · Decision Letter 1]

20 Aug 2021

May predator body-size hamper furtive predation strategy by aphidophagous insects?

PONE-D-21-16778R1

Dear Dr. Meseguer,

We’re pleased to inform you that your manuscript has been judged scientifically suitable for publication (BUT, please see and apply ADDITIONAL EDITOR COMMENTS below) and will be formally accepted for publication once it meets all outstanding technical requirements.

Kind regards,

Ramzi Mansour

Academic Editor

PLOS ONE

Additional Editor Comments:

The following revisions should be made by the authors on the PROOFS of their accepted article:

L43:  replace "of aphid pest"  with  "of aphids"

L51:  replace "phymatidae"   with  "phymatids"

L54: replace "chamaemiid"   with  "chamaemyiid"

L90:  replace  "(1)"   with  "(i)"

L92:  replace "(2)"   with  "(ii)"

L95:  replace  "(1)"   with  "(i)"

L97:  add the authorship "(Pallas)"   after  "Harmonia axyridis" considering this is the first mention of this species after the Abstract

L98:  replace "(2)"  with  "(ii)"

L120:  replace "The Asian ladybird"   with  "The harlequin ladybird"

L152:  the aphid-infested leaf,

L156:  delete "too"

L163:  replace  "a control"   with  "A control"

L163:  delete the " . "  after  "pedators"

L182:   from 13 to 16 individuals,

L185:  replace  "(1)"   with  "(i)"

L186:  replace "(2)"  with  "(ii)"

L256:   were those that elicited

L277:  for consistency with L163,  "Control"  should not start with a capital letter "C" but with "c" and should not be italicized

L285:  replace the comma after "L1", "L2"  and "L3"  with " : "

L416:  replace "the Asian ladybird beetle"   with  "the harlequin ladybird"

L430:   have a furtive predatory behavior; and whether the
---

## [Editor Report · Acceptance letter]

24 Aug 2021

PONE-D-21-16778R1 

May predator body-size hamper furtive predation strategy by aphidophagous insects? 

Dear Dr. Meseguer:

I'm pleased to inform you that your manuscript has been deemed suitable for publication in PLOS ONE. Congratulations! Your manuscript is now with our production department. 

Kind regards, 

on behalf of

Dr. Ramzi Mansour 

Academic Editor

PLOS ONE